# OCCUPY & SPECIFY: INVESTIGATIONS INTO A MAXIMUM CREDIT ASSIGNMENT OCCUPANCY OBJECTIVE FOR DATA-EFFICIENT REINFORCEMENT LEARNING

## ABSTRACT

The capability to widely sample the state and action spaces is a key ingredient toward building effective reinforcement learning algorithms. The trade-off between exploration and exploitation generally requires the use of a data model, from which novelty bonuses are estimated and used to bias the return toward wider exploration. Surprisingly, little is known about the optimization objective followed when novelty (or entropy) bonuses are considered. Following the "probability matching" principle, we interpret here returns (cumulative rewards) as set points that fixate the occupancy of the state space, that is the frequency at which the different states are expected to be visited during trials. The circular dependence of the rewards sampling on the occupancy/policy makes it difficult to evaluate. We provide here a variational formulation for the matching objective, named MaCAO (Maximal Credit Assignment Occupancy) that interprets rewards as a log-likelihood on occupancy, that operate anticausally from the effects toward the causes. It is, broadly speaking, an estimation of the contribution of a state toward reaching a (future) goal. It is constructed so as to provide better convergence guaranties, with a complementary term serving as a regularizer, that, in principle, may reduce the greediness. In the absence of an explicit objective occupancy, a uniform prior is used, making the regularizer consistent with a MaxEnt (Maximum Entropy) objective on states. Optimizing the entropy on states in known to be more tricky than optimizing the entropy on actions, because of an external sampling through the (unknown) environment, that prevents the propagation of a gradient. In our practical implementations, the MaxEnt regularizer is interpreted as a TD-error rather than a reward, making it possible to define an update in both the discrete and continuous cases. It is implemented on an actor-critic off-policy setup with a replay buffer, using gradient descent on a multi-layered neural network, and shown to provide significant increase in the sampling efficacy, that reflects in a reduced training time and higher returns on a set of classical motor learning benchmarks, in both the dense and the sparse rewards cases.

## 1 PROBLEM STATEMENT

Learning in the real world implies dealing with very large, potentially unlimited environments, over which the data to collect is seemingly infinite. *Efficient* exploration is thus one of the key aspects of open-ended learning Santucci et al. (2020), when no final model of the environment can feasibly be expected to be engineered or trained. On the one side, having access to unlimited data is very beneficial for the training of complex multi-layered perceptrons, for they are known to rely on large datasets to improve their performance. On the other side, the circular dependence between the learning algorithm and the data on which it operates renders the learning very tricky, at high risk of data overfitting and trapping in local optima. The open-ended learning problem is generally addressed through the lens of the reinforcement learning framework (Sutton et al., 1998), where rewards are collected during the interaction, and the selection of action is fit so as to maximize the total number of positive rewards, and prevent the encounter of negative ones. Fitting behaviour to rewards is however at the risk of ignoring important data from the rest of the environment, where putatively more rewarding regions may be neglected. The agreement of reward-seeking (that is exploitation) with data collecting (that is exploration), is still one of the fundamental issues of modern artificial intelligence.

An important effort has recently been put on reframing the reinforcement learning setup into a more general probabilistic inference framework, allowing to link rewards seeking and data modelling under a single perspective (Furmston & Barber, 2010; Levine, 2018; Haarnoja et al., 2018; Abdolmaleki et al., 2018; Fellows et al., 2019). This greater focus over the data collection problem is linked to an important set of training algorithms, that contain some forms of exploration bonuses, including "curiosity" drives (Schmidhuber et al., 2009; Pathak et al., 2017), intrinsic rewards (Oudeyer et al., 2007) and pseudo-counts (Bellemare et al., 2016; Tang et al., 2017). However, at the difference of the classic optimization on rewards alone, where the Bellman optimum is well defined, there is still no consensus about the objective followed when optimizing both on rewards and data collection under a variational perspective (Eysenbach & Levine, 2019). The data collection problem is effectively shadowed by the reward maximization objective, under which it is still considered as a incidental component. An important body of work has recently been devoted to addressing the data collection problem as such, with the notable design of the MaxEnt algorithm (Hazan et al., 2019), State Marginal Matching (Lee et al., 2019) and E3D (Daucé, 2020), that aim at fitting the distribution of the states encountered to a uniform distribution, in the absence of definite rewards. This is here referred as a MaxEnt-on-state principle (or MaxEnt to be short), not to be confounded with the MaxEnt-on-actions principle implemented in the soft actor critic (Haarnoja et al., 2018) for instance. Following a MaxEnt objective means optimizing the policy so as the states visited are maximally variable, ideally uniformly visiting all possible states. We develop in the following a possible extension of the MaxEnt principle, that brings a considerable simplification in the expression of the evidence lower bound (ELBO) with regards with the existing literature (Furmston & Barber, 2010; Abdolmaleki et al., 2018; Fellows et al., 2019). In contrast to pure MaxEnt, our approach provides a way to combine the MaxEnt objective with a reward maximization objective, under a variational inference perspective. An intriguing property of the resulting ELBO formula is that the future states (the ones that are visited after the current observation) play the role of a *model* for the current data, participating in the elaboration of the returns collected under the current policy. This gives ways toward optimizing the policy with respect to the distribution of the data, and provides a principled justification to the use of intrinsic rewards in the design of reinforcement learning algorithms.

## 2 PRINCIPLES

### 2.1 PROBABILITY MATCHING RL

We assume an agent acting in a fully observable environment. The state of the environment is provided by an observation $s \in \mathcal{S}$, with $\mathcal{S}$ the set of all possible states. The agent can act on the environment through its actuators. Such a motor command is described by $a \in \mathcal{A}$, with $\mathcal{A}$ the set of all motor commands. In the following, the capital letters $S$ and $A$ will reflect random variables on $\mathcal{S}$ and $\mathcal{A}$, and the lower cases $s$ and $a$ will either reflect observations or random draw realizations. The decision of which action to choose relies on a *policy*, that maps the current observation to the action space, generally expressed in a conditional probabilistic form $\pi(a|s)$.

A reinforcement learning problem consists in finding a policy $\pi^*$ that maximizes a certain objective function, without knowing the physical or mechanical properties of the environment. It is supposed here, for simplicity, that the dynamics of the environment is Markovian (no hidden states). Moreover, the environment is providing an auxiliary signal called the reward. Sending an action to the environment makes it possible to access to a new state $s'$, and to obtain a reward $r \in \mathbb{R}$. A classic objective in learning is to maximize the global return, generally described as a discounted sum of future rewards over all possible trajectories. Let us now denote by $s_t$ the state visited at time $t$ and $\tau(s_t) = (s_{t+1}, ..., s_{t+T}, ...)$ a certain pathway that is visited after observing $s_t$. During this visit, a certain number of rewards can be collected, and $R(\tau)$ is the (discounted) return obtained over $\tau$, i.e. $R(\tau) = \sum_t \gamma^t r_t$, with $\gamma \in [0, 1[$ a discounting factor that sums up the rewards up to an "horizon" of the order of $\frac{1}{1-\gamma}$. This said, the dynamic programming objective (Bellman, 1966), is the result of $\pi^* = \max_\Pi \mathbb{E}_{s \sim p(S_0), \tau \sim p_\pi(\tau|s)} R(\tau)$, with $p(S_0)$ the distribution of initial states, and $\Pi$ the set of all conditional policies. When a state transition model $p(S'|s, a)$ is provided, the unique solution is given by the dynamic programming recurrent equation in the discrete case (Bellman, 1966). On the contrary, a large panel of reinforcement learning techniques allow to approach the solution in the model-free setup, assuming an effective sampling of all state-action pairs (Sutton et al., 1998).

We are here interested in a different class of objective function, that rely on fitting the rewards toward probabilities of state *occupancies*. A reward should indicate in which proportion the different states (and actions) should be visited (and selected) during trials (with the idea that the states providing high return should be visited more often than the ones providing low returns). Solving the reinforcement learning problem then means to match the external cue to an actual distribution of visit over states and actions, where a differential in rewards only indicates a difference in the number of visits, allowing to seek rewards in a flexible way (so it is also referred as to "soft" reinforcement learning (Haarnoja et al., 2018)). This idea stems back from empirical observations on human and animal behaviors, and was coined the "matching law" in the operant conditioning literature (Herrnstein, 1961; Eysenbach & Levine, 2019).

## 2.2 STATE OCCUPANCY AND CONDITIONAL STATE OCCUPANCY

Matching rewards to probabilities can be done in many different ways. We frame here the probability matching reinforcement learning problem into a *state occupancy matching problem*. It relies on the use of an occupancy distribution, that is a density of state visit under a certain policy. Importantly, it ignores the time order at which the different states are visited, still conserving some aspects of causality between states in the form of conditional probabilities as we see later. Dating back from Dayan (1993), an occupancy distribution is a distribution on states, designed so as to match with the distribution measured over the trajectories of the MDP.

Following the definitions of (Puterman, 2014; Ho & Ermon, 2016; Hazan et al., 2019), a gamma-absorbing state occupancy of a Markov Decision process (with a policy $\pi$) is the (discounted) density of visit of the states — or (state, action) pairs — of the environment when starting from the initial distribution $p(S_0)$. It is defined, as:

$$\begin{cases} \rho_\pi(s) = (1 - \gamma)p_0(s) + \gamma \sum_{s',a'} p(s|s', a')\pi(a'|s')\rho_\pi(s') \\ \rho_\pi(s, a) = \pi(a|s)\rho_\pi(s) \end{cases} \tag{1}$$

so that any policy $\pi$ settled on an MDP defines an occupancy on the states of that MDP. It comes that, inversely, any valid (state, action) occupancy (meaning that this occupancy is effectively feasible in a given agent/environment setup), defines a unique corresponding policy:

$$\pi(a|s) = \frac{\rho(s, a)}{\rho(s)} \tag{2}$$

that is a softmax (stochastic) conditional policy over the states.

Following the same reasoning, let $\rho_\pi(S^+|s...)$ the *conditional occupancy* be defined recursively. Let $\mathcal{T}_\pi(s)$ the set of trajectories starting from $s$:

$$\forall s^+ \in \mathcal{T}_\pi(s), \rho_\pi(s^+|s...) = p_\pi(s^+|s) + \gamma \sum_{s' \in \mathcal{T}_\pi(s)} p_\pi(s^+|s')\rho_\pi(s'|s...)$$

The triple dots (...) are intended to help distinguish the one-step distribution $p_\pi(S'|s)$ from the long-term distribution $\rho_\pi(S^+|s...)$. This conditional distribution provides a description of the "future" of $s$, that is the distribution of states that will most probably follow $s$. It can be seen as an instance of the "successor" representation of states initially proposed by Dayan (1993). Those future states will generally be noted $s^+$, with the '+' exponent meaning the state being measured "further away in time".

## 2.3 MATCHING REWARDS TO OCCUPANCIES

Those definitions provide a way toward interpreting rewards as occupancy templates, allowing to implement the "matching law" in a principled way. The mapping of rewards toward probabilities relies on using exponentiated returns in the parameters of a stochastic policy, such as in the softmax (or Boltzmann) decision rule case. Let $\pi(a|s) = \frac{\exp \beta Q(s,a)}{K(s)}$ with $K(s) = \sum_a \exp \beta Q(s, a)$, with $\beta$ the "inverse temperature", and the state-action value $Q(s, a)$ representing the total return estimated at $(s, a)$.

Let $\tau = (s_0, s_1, ..., s_t, ...)$ a certain trajectory observed on the MDP under the policy $\pi$. The set of all possible trajectories is noted $\mathcal{T}$, $p_\pi(\tau)$ is a measure over the trajectories for a certain policy

$\pi$, and $\rho_\pi$ is the corresponding occupancy on *states*. Consider for instance the series of rewards encountered when following $\tau$. It comes that:

$$\mathbb{E}_{s_0 \sim p_0} V(s_0) = \mathbb{E}_{\tau \sim p_\pi(\mathcal{T})} \sum_t \gamma^t r(s_t, a_t) \approx \mathbb{E}_{\substack{s \sim \rho_\pi(S) \\ a \sim \pi(A|s)}} r(s, a) \sum_t \gamma^t = \mathbb{E}_{s, a \sim \rho_\pi(S,A)} \frac{r(s, a)}{1 - \gamma}$$

so that $\forall t$, $\frac{r(s_t, a_t)}{1 - \gamma}$ is interpreted as an estimator of $V(s_0)$.

Next, for any $t$, it comes that $\forall t' > t$, $\frac{r(s_{t'}, a_{t'})}{1 - \gamma}$ is an estimator of the state-action value $Q(s_t, a_t)$, i.e:

$$Q(s_t, a_t) = \mathbb{E}_{\substack{\tau \sim p_\pi(\mathcal{T}) \\ s_t \in \tau}} \sum_{t' > t} \gamma^{(t'-t)} r(s_{t'}, a_{t'}) \approx \mathbb{E}_{\substack{s^+ \sim \rho_\pi(S^+|s_t, a_t \dots) \\ a^+ \sim \pi(A|s^+)}} \frac{r(s^+, a^+)}{1 - \gamma} \tag{3}$$

In that setup, the rewards are interpreted as value samples. This means, in short, that each future reward $r(s^+, a^+)$ takes the role of a "model" for the total return $Q(s_t, a_t)$. The models are weighted according to the conditional occupancy $\rho_\pi(S^+, A^+|s, a\dots)$, that takes the role of the "mixture".

Then, noting that $\log \pi(a|s) = \beta Q(s, a) - K(s)$, we define :

$$\bar{R}(s, s^+, a^+) \triangleq \frac{r(s^+, a^+)}{1 - \gamma} - \frac{1}{\beta}(K(s) - \log \rho_\pi(s)) \tag{4}$$

said the "extended" return composed of the return estimator plus a virtual baseline. Then, because the policy and the occupancy are exchangeable from eq.(2), each reward collected after $(s, a)$ may also take the role of a "model" for the occupancy, sampled from the conditional occupancy, i.e.:

$$\log \rho_\pi(s, a) \approx \mathbb{E}_{\substack{s^+ \sim \rho_\pi(S^+|s, a \dots) \\ a^+ \sim \pi(A|s^+)}} \beta \bar{R}(s, s^+, a^+) \tag{5}$$

## 2.4 DENSITY MATCHING OPTIMIZATION

Assuming $\pi$ a current policy, $\rho_\pi$ a corresponding occupancy, and taking $\langle \widetilde{\log \rho}(s, a) \rangle_{A^+, S^+, \pi}$ as a shorthand for $\mathbb{E}_{\substack{s^+ \sim \rho_\pi(S^+|s_t, a_t \dots) \\ a^+ \sim \pi(A|s^+)}} \beta \bar{R}(s, s^+, a^+)$ the sampling-based optimization writes:

$$\pi^* = \underset{\Pi}{\operatorname{argmax}} \, \mathbb{E}_{s, a \sim \rho_\pi(S,A)} \langle \widetilde{\log \rho}(s, a) \rangle_{A^+, S^+, \pi} \tag{6}$$

which is a cross-entropy objective that aims at fitting $\rho_\pi(S, A)$ with $\langle \widetilde{\rho}(S, A) \rangle$. The optimization can be done, for instance, by optimizing a current policy $\pi$ by stochastic gradient ascent on the objective, which conducts to maximizing the return, under a softmax policy, like in classic policy gradient.

However, we can make a step further by trying to estimate how far is $\langle \widetilde{\log \rho}(s, a) \rangle$ from the optimum $\log \rho^*(s, a)$. Interestingly, for any distribution on the future states $q(S^+, A^+)$, the following inequality holds:

$$\mathbb{E}_{s^+, a^+ \sim q(S^+, A^+)} \beta \bar{R}(s, s^+, a^+) \geq \mathbb{E}_{s^+, a^+ \sim q(S^+, A^+)} \beta \bar{R}(s, s^+, a^+) - \mathcal{D}_{\mathbf{KL}}(q(S^+, A^+)||\rho^*(S^+, A^+|s, a\dots))$$

$$\approx \mathbb{E}_{s^+, a^+ \sim q(S^+, A^+)} \log \rho^*(s, a) - \mathcal{D}_{\mathbf{KL}}(q(S^+, A^+)||\rho^*(S^+, A^+|s, a\dots)) \tag{7}$$

providing a variational (log-) evidence lower bound (ELBO) interpretation of the maximization on $\beta \bar{R}(s, s^+, a^+)$. At the convergence of the gradient ascent, the distribution $q$ is expected to match the posterior $\rho^*$, and the inequality would becomes an equality, i.e.

$$\log \rho^*(s, a) = \mathbb{E}_{s^+, a^+ \sim \rho^*(S^+, A^+|s, a\dots)} \beta \bar{R}(s, s^+, a^+) - \mathcal{D}_{\mathbf{KL}}(\rho^*(S^+, A^+|s, a\dots)||\rho^*(S^+, A^+|s, a\dots))$$

This formula (7) is not directly usable in optimization, because the optimal posterior $\rho^*$ is not specified, but provides new hints into interpreting the current occupancy at the light of its future effects. The actual optimization is indeed done on $\bar{R}(s, s^+, a^+)$ solely, irrespective of the divergence bias. This bias is expected to fade away with the progress of the training, making the sampling of the rewards more and more accurate at estimating the parameters of the policy/occupancy. This however, is not guaranteed, and the optimization on the (pseudo) cross entropy is at risk of keeping a high divergence throughout the optimization, hindering the convergence toward the optimum. This illustrates a more general problem that is the lack of efficacy in sampling the data (the posterior occupancy), a more robust upper bound guarantee would be preferred, even at the risk of a lesser final optimality with regard to the Bellman optimum.

## 2.5 Maximal Credit Assignment Occupancy

We now introduce the main insight of our Maximal Credit Assignment Occupancy (MaCAO) model. It appears that the formulas (5) and (6) are not entirely satisfactory: the average over the rewards translates into an average over log-density templates, that actually depend on $(s^+, a^+)$, though it is not explicit in the formula. Expressing this dependency is possible by reshaping the ELBO formula (7) like :

$$\log \rho^*(s, a) \geq \mathbb{E}_{s^+, a^+ \sim q(S^+, A^+)} \log \rho^*(s, a|...s^+, a^+) - \mathcal{D}_{\mathbf{KL}}(q(S^+, A^+)||\rho^*(S^+, A^+)) \quad (8)$$

This introduces a new conditional distribution, namely $\rho(S, A|...s^+, a^+)$, that is the frequency at which $(s, a)$ may precede $(s^+, a^+)$ in the iteration of the dynamics. This distribution expresses an "anti-causal" relationship between the future states and the current observations, that is the exact measure of how much $(s, a)$ is instrumental in reaching $(s^+, a^+)$. This is also said the "credit assignment" in the reinforcement learning literature (Sutton, 1988; Harutyunyan et al., 2019). Here $\rho^*(s, a|...s^+, a^+)$ represents the target credit assignment. From the Bayesian perspective, it is interpreted as a log-likelihood of the "data" $(s, a)$, given the "model" $(s^+, a^+)$, that is a way to say that $(s^+, a^+)$ exerts a control on $(s, a)$.

This conducts to reconsider the rewards as log-likelihood templates rather than occupancy templates. It comes from lemma 4 (see appendix)) that: $\mathbb{E}_{s^+, a^+ \sim q(S^+, A^+)} \log \rho^*(s, a|...s^+, a^+) \geq \mathbb{E}_{s^+, a^+ \sim q(S^+, A^+)} \bar{R}(s, s^+, a^+)$, so that:

$$\log \rho^*(s, a) \geq \mathbb{E}_{s^+, a^+ \sim q(S^+, A^+)} \bar{R}(s, s^+, a^+) - \mathcal{D}_{\mathbf{KL}}(q(S^+, A^+)||\rho^*(S^+, A^+)) \quad (9)$$

This new objective provides a variational Bayesian perspective on the density matching optimization, with $q$ (the conditional occupancy) taking the role of the variational distribution. This is equivalent to augmenting the return with a supplementary divergence term on the conditional occupancy. Then, matching $\log \rho^*(s, a)$ with the returns becomes identical to maximize the ELBO (8) from variational inference.

The loss is composed of two complementary terms, a first term being the occupancy matching on cumulative rewards (that is consistent with the softmax optimization), while the second term is the explicit density matching of a posterior with a prior. The prior takes the role of a supervision signal, that aims at putting a constraint on the conditional occupancy (that represents the exploration pattern). By construction of the loss, the right term is made to shape the posterior occupancy, that conducts the evaluation of the return. This is formally analog to the case of Bayesian inference where the prior serves as a regularizer that tends to counteract the overfitting of the data. This sort of regularizer is highly expected in reinforcement learning that is known to be prone to overfitting. The analogy with Bayesian inference suggests for instance to consider a Gaussian or a *uniform* distribution. This has important consequences though. Taking a prior that is not the target occupancy strongly modifies the interpretation of the loss, that is now composed of two *concurrent* terms. The likelihood part aims at fitting the occupancy with the rewards collected, while the divergence part aims at fitting the posterior occupancy with an arbitrary prior. This breaks the original symmetry, for the implies to concurrently follow two different objectives. For instance, the role of regularizer that would be devoted to a uniform prior introduces a bias in the probability matching, for the target occupancy is not anymore the softmax-Bellman optimum, but rather an intermediary occupancy that combines the Bellman optimum and a uniform occupancy. This is expected to be beneficial for a wider exploration, helping to avoid overfitting, but this is at the cost of a relaxed constraint on fitting to the optimum.

This expression of a variational upper bound is reminiscent of the standard variational bounds considered in the reinforcement learning literature (Furmston & Barber, 2010; Abdolmaleki et al., 2018), from which it could be interpreted as an occupancy-oriented variant. By breaking the temporal chain dependence, however, it surprisingly provides a justification for considering an entropy *on states* in the optimization of the policy. Indeed, the classic variational optimization operates on chained sequences of observations, for which the state transitions vanish during the optimization of $\pi$ (Haarnoja et al., 2018). In our case, ignoring the sequential time order establishes both the future states and the future actions as parameters of the policy, over which it should be optimized. This allows to address the data efficiency through explicitly optimizing on the exploration path. It moreover provides a room to the maxent-on-states term (Hazan et al., 2019), as a regularization incentive in a more general expression. This conceptual shift has however an important consequence on the objective followed, for the target occupancy is not anymore equal to the softmax Bellman optimum.

## 3   METHOD

The main ingredients for an efficient implementation is the access to a wide variety of samples $(s, a, r, s', a')$ over which optimization can be carried out on parameterized policy $\pi_\theta$ (said the "actor") and a parameterized action value function $Q_\psi$ (said the "critic"). It is here implicitly assumed that both the actor and the critic consist of multi-layered perceptrons, containing many parameters and organized in layered weights, over which a gradient descent is operated on losses expressed as negative objectives.

Assuming an off-policy approach, we consider a replay buffer containing many samples of states, actions and rewards as observed from interacting with the environment. In a variational setup, one can assume an alternation between two complementary steps. A first step, said the "estimation" step, consists in evaluating a distribution over the parameters of the model, that is here assumed to be a distribution over future states, identified as the conditional occupancy under the current policy $q_\pi$. This distribution is then exploited in a second step, said the "maximization" step, where the parameters of the policy/Q-function are updated so as to maximize the proximal objective (that is fitting a policy to a distribution of returns obtained from the current occupancy).

We assume here the estimation of $q_\pi$ being obtained from a parametric or non-parametric method over a sufficiently large sample of recent states (or state, action pairs), and concentrate on the optimization of the actor under the guidance of the critic. Knowing $q_\pi$, and given a sample $(s, a, r, s', a')$, an element of optimization is given by the log-difference $\log q_\pi(s', a') - \log \rho^*(s', a')$ that is a point estimate of the Kullback-Leibler divergence, that is expected to be minimized during the optimization of the policy. This term is supposed to be differentiated with regards to $\pi$, providing a first gradient direction that should contribute to improving the policy toward a wider occupancy on future states. A second and independent element of optimization is the parameterized action-value mapping $Q(s, a)$, relying on the TD-error construct Sutton (1988), based on approximating the future rewards with the best current estimate at $t + 1$, defining a gamma-discounted proximal objective value $\tilde{Q}(s, a) = r + \gamma Q(s', a')$. The squared difference $\lambda(\tilde{Q}(s, a) - Q_\psi(s, a))^2$, with $\lambda$ a precision hyperparameter, is known as the mean-squared Bellman error (MSBE), providing a second gradient that aims at maximizing the return with regards to the policy parameters. The concurrence of both gradients contains the necessary elements to combine exploration and exploitation in an principled way.

Building a full parametric model of the occupancy is however a difficult task that should be undertaken with care. The building of such a probabilistic model is indeed at a non negligible cost of regressing parametric distributions from samples, that inherently contain design choices and a specific optimization on a set of latent parameters (like in the case, e.g., of auto-encoders). For the sake of simplicity, we consider here the case of a non-parametric estimator of the occupancy distribution $q_\pi$. This could appear counter-intuitive at first glance, for the update of the policy is supposed to rely on backpropagating gradients through the estimator. As a workaround, we provide here a method allowing to directly inject the gradient information in the design of the $Q$-function.

Assume a parameterize $Q$-function $Q_\psi$ that should undergo a dual optimization under two concurrent objectives. This implies in short that the value settled in the $Q$-function may arbitrate between the reward-seeking and the occupancy-seeking objectives, so that maximizing $\pi$ with respect to $Q$ only may be equivalent to the previous concurrent optimization. Considering $(s, a, r, s', a')$ a sample, a simple way to retain the divergence information is to simply add the log difference information $\log q_\pi(s', a') - \log \rho^*(s', a')$ to the current $Q$-function, making it possible to recover the original formula. This sketch of idea implies the use of two concurrent update rules in the $Q$-function itself. A first term $L_{\text{ref}}(s, a, r, s', a')$ would be the traditional MSBE loss on rewards. The second loss needs to consider the log-difference itself as an *error*, for it to be "contained" (so to say) in $Q$ after the update. Consider the implicit reward :

$$r_{\text{KL}}(s, a) \triangleq (1 - \gamma) \left[ Q_\psi(s, a) - \frac{1}{\beta} (\log q_\pi(s', a') - \log \rho^*(s', a')) \right] \tag{10}$$

and let $\hat{Q}(s, a) = r_{\text{KL}}(s, a) + \gamma Q_\psi(s', a')$. Then a complementary MSBE loss is:

$$L_{\text{KL}}(s, a, r, s', a') = (r_{\text{KL}}(s, a) + \gamma Q_\psi(s', a') - Q_\psi(s, a))^2$$

---

**Algorithm 1** Maximum Credit Assignment Occupancy (MaCAO)

---

**Require:** $\pi_\theta$ (actor), $Q_\psi$ (critic), $\mathcal{B}$ (replay buffer), $\beta, \gamma, \lambda$ (hyperparameters)
  **while** number of trials not exceeded **do**
    initialize the environment
    **while** trial not terminated **do**
      observe $s$
      choose $a \sim \pi_\theta(A|s)$
      read $s', r$
      store $(s, a, r, s')$ in $\mathcal{B}$
    **end while**
    **if** $\mathcal{B}$ is full enough **then**
      randomly sample a batch of (next) states $\{s^+, ...\}$ from $\mathcal{B}$.
      estimate $q$ with a nonparametric method.
      **while** number of batch updates not exceeded **do**
        randomly sample a batch of transitions $b = \{(s, a, r, s'), ...\}$ from $\mathcal{B}$.
        **for** all $(s, a, r, s') \in b$ **do**
          estimate $\log q(s')$ and set $r_{\text{KL}}$ (eq. 10)
          sample $a' \sim \pi_\theta(A|s')$
          calculate $L_{\text{MaCAO}}(s, a, r, s', a')$ (eq. 11)
        **end for**
        update the critic $Q_\psi$ by gradient descent over all losses.
        **for** all $s \in b$ **do**
          sample $a \sim \pi_\theta(A|s)$
          estimate $Q_\psi(s, a)$
          calculate $L_{\text{act}} = -Q_\psi(s, a) + \frac{1}{\beta} \log \pi_\theta(A|s)$ (including the reparameterization trick)
        **end for**
        update the actor $\pi_\theta$ by gradient descent over all losses.
      **end while**
    **end if**
  **end while**

---

So that the final loss expression is:

$$L_{\text{MaCAO}}(s, a, r, s', a') = \lambda \left( r + \gamma Q_\psi(s', a') - Q_\psi(s, a) \right)^2 \tag{11}$$
$$+ \left( \left[ -\frac{(1-\gamma)}{\beta} (\log q_\pi(s', a') - \log \rho^*(s', a')) \right] + \gamma \left( Q_\psi(s', a') - Q_\psi(s, a) \right) \right)^2$$

The additional *precision* hyperparameter $\lambda$ plays here the role of an extra regularizer, helping to adjust between both terms in case of highly sparse rewards. The main lines of our implementation are provided in algorithm 1 (see Appendix), that fits the pursuit of the MaCAO objective in an actor-critic setup. It relies on a wide use replay buffers (Mnih et al., 2013) to regularize the gradient over batches that mix the samples from many different trials. From this perspective, an important shortcut is our on-the-fly calculation of the log-occupancy, with the help of kernel-based density estimation method (Pedregosa et al., 2011) from an initial sampling of (future) states from the buffer at each start of an update sequence (line 10). This occupancy sample remains quite limited in number (about 1000) in order to avoid unnecessary computer overload.

## 4 RESULTS

In order to reach state-of-the art efficacy, many algorithmic improvements need to be included in supplement to the baseline algorithm[1]. This concerns in particular the use of target Q-networks updated at slower pace (Mnih et al., 2013), and the clipped double-Q trick (Fujimoto et al., 2018). Our implementation is moreover drawn over the "spinning-up" open source framework (Achiam, 2018), allowing for a direct comparison with the state of the art. We consider here for comparison the soft actor-critic method (SAC) (Haarnoja et al., 2018), proximal policy optimization (Schulman et al., 2017) and TD3 (Fujimoto et al., 2018). Our method was tested over several benchmark environments, as provided by the "Gym" suite (Brockman et al., 2016). We concentrate here on the continuous states and actions case, that is the most challenging one with regards to function approximation.

---

[1]code freely available at http://github.com/xxx/yyy (to come).

The different setups are compared on the basis of the returns collected during training. This is expressed as average return (that is the total sum of rewards gathered at the end of an episode), the average reward (total rewards collected divided by the episode length) and cumulative rewards (the total sum of rewards collected at a given stage of the training). The width of the occupancy over the state space is not compared here, for the other frameworks are not designed to optimize it. The different environments differ in scale, difficulty and rewards density. All continuous problems proposed in the library provide dense rewards, that are a compound of negatively and positively weighted extrinsic informations, like the energy consumption, the speed of the agent or its elevation. The problems separate in two broad categories. A first class of problems provides only dense rewards. A second class of problems have, in addition, a supplementary sparse reward taking the form of an "end-of-episode" bonus or penalty. In that case, the dense rewards may (or may not) contain relevant information with regards to the task at hand.

From that prospect, the most unfavorable problem is the Continuous Mountain Car problem (first row of figure 4). Here the dense rewards only refer to the energy consumption, at the exception of a +100 end-of-episode bonus obtained at the hilltop. This inevitably conducts baseline algorithms to remain stucked at the bottom of the hill, where the energy consumption is low. Only our

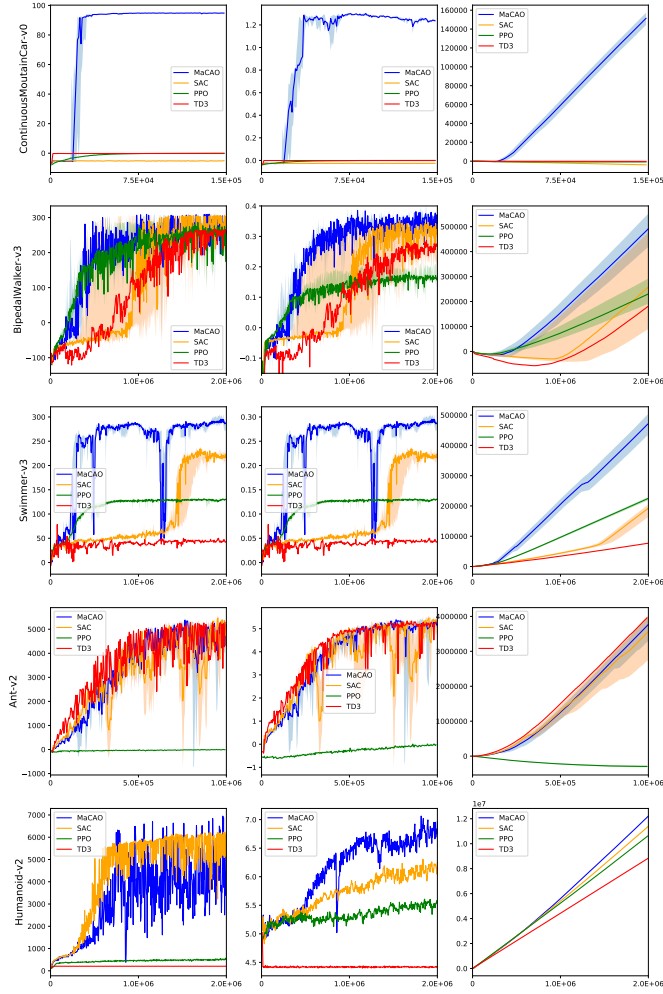

Figure 1: **Methods comparison** Average episode rewards, average rewards and cumulative rewards are compared in the course of learning for the MaCAO, SAC, PPO and TD3 frameworks, on 5 continuous state/continuous control problems. Row 1: Gym *Continuous Mountain Car* problem. $\beta = 10, \lambda = 0.1, \gamma = 0.99$, 2 hidden layers with $N = 32$ neurons. 5 seeds. Row 2: Gym/MuJoCo *Swimmer* problem. $\beta = 30, \lambda = 0.3, \gamma = 0.995$, 2 hidden layers with $N = 32$ neurons. 5 seeds. Row 3: Gym/Box2D *Bipedal Walker*. $\beta = 30, \lambda = 1, \gamma = 0.99$, 2 hidden layers with $N = 64$ neurons. 5 seeds. Row 4: Gym/MuJoCo *Ant*. $\beta = 10, \lambda = 0.3, \gamma = 0.99$, 2 hidden layers with $N = 256$ neurons. 5 seeds. Row 5: Gym/MuJoCo *Humanoid*. $\beta = 10, \lambda = 3, \gamma = 0.98$, 2 hidden layers with $N = 256$ neurons. 1 seed.

approach, that contains an explicit incentive for widening the occupancy of the state space, has the capability to reach the most rewarding states, finally providing a policy that solves the task.

The Bipedal Walker (second row of figure 4) is also a problem that combines dense and sparse rewards. A negative (-100) reward is undergone when the agent falls down, and a positive (+100) reward is gained when the agent reaches the end of the track. The continual dense rewards provide an incentive for staying upright and increase the velocity. This task reveals more tricky to train than expected, and contains enough variability for the agent to develop various gaits and locomotion patterns over the course of learning. Our approach shows here a clear advantage, that is maybe more obvious when comparing with the SAC. Like in the Mountain car, the problem is about reaching a final (distal) end-of-path objective, from which a strong bonus allows to "freeze" the behavior in a favorable locomotion pattern. The (S-shaped) discontinuity in the MaCAO and the SAC learning

curves reflect the reaching of the distal objective, after which close-by policies are followed. The difference in the two curves is the *time* at which it is attained, that is less than $10^5$ iterations in the first case, and more than $10^6$ in the second. This one order difference illustrates the disadvantage of occupancy-agnostic optimization methods.

The swimmer task (third row) is concerned with the development of a locomotion pattern that is swimming in a liquid medium. The reward is only the speed at which an eel-like agent manage to swim over the place (that is coordinating segments in a periodic manner). This tasks contain a local optimum that corresponds to a rower pattern that coordinates the extremal segments, and a global optimum that corresponds to a classic swimming ripple from the head toward the tail. Despite its apparent simplicity, only an extensive exploration such as the one provided by our approach allows to reach the optimum.

The fourth task, known as the "ant" aims at controlling the locomotion of a 4-legged agent. The state space contains a detailed account of joint angle and torque moments plus contact sensors in a 111 dimension observation vector (Schulman et al., 2015), but the control space is more reduced (8 DOFs). Here again the displacement speed is the main incentive, with a survival bonus, and an energy cost penalty. All 3 actor critic frameworks (namely MaCAO, SAC and E3D) are here capable to reach a decent locomotion pattern in about 400000 iterations of the dynamics, which can be considered data efficient here. No clear advantage is found here for our approach.

Last, the humanoid task shows a large number of degrees of freedom, and the unlimited number of possible locomotion patterns often result in strange-looking final gaits. Only the SAC and the MaCAO methods allow here to reach decent locomotion patterns in the limited number of steps considered. When looking in detail, the light advantage observed for the SAC algorithm on the average episode return is reversed when considering the average return. This apparent contradiction is explained when looking at the detailed behavior. Here, the high-speed risky locomotion patterns developed in the MaCAO framework result in a higher number of early failures. This is not related to a risk-seeking incentive, but is rather explained by a tendency to maintain a high diversity of behavior while pursuing the reward-guided objective, which reveals to be more risky when the balance of the body needs to be maintained over time.

## 5   DISCUSSION

This work participates to a general trend toward the development of data models in reinforcement learning, that provide ways to help the agent toward better exploring the world. This is known of practical use and has been largely exploited so far in the large family of curiosity-driven and maximum-entropy algorithms. Our contribution here is to provide a more detailed appraisal of the benefits and putative risks of such a construct. It is shown here to frame into a larger Bayesian/variational optimization where the future data plays the role of a model, and where an evidence lower bound is maximized through gradient ascent over the policy parameters. The general principles exposed point to the importance of an occupancy model that synthesizes the general distribution of the agent's environmental states over which it can act (defining a virtual "territory"). Those occupancy models are the subject of frequent updates as the exploration progresses and that new states are undisclosed during the course of the training. By making an additional uniform prior assumption on the occupancy, the resulting loss expresses a balance between two concurrent tendencies, namely the widening of the occupancy space and the maximization of the rewards, reminding of the classical exploration/exploitation trade-off. The consequence is a shift in the target occupancy pursued, that relaxes the constraint on fitting the initial Bellman objective. Both are embodied in a MSBE Loss operating on a single Q-function in our implementation (though this is not necessary the case). Computer simulations illustrate the benefit of our conceptual developments, both in the case of sparse and dense rewards.

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

## A  MATHEMATICAL COMPLEMENTS

### A.1  LOWER BOUNDS

Consider a triplet of random variables $(S, A, Z)$ defined on the domains $\mathcal{S} \times \mathcal{A} \times \mathcal{Z}$ and obeying to the mixture probability $p(S, A, Z)$. Les $s \in \mathcal{S}$ be a "state", $a \in \mathcal{A}$ be an "action" and $z \in \mathcal{Z}$ be a "successor". Define $p(S)$, $p(A)$, $p(Z)$, $p(S, A)$, $p(S, Z)$ and $p(A, Z)$ the corresponding marginals and define any conditional $p(X|Y) = \frac{p(X,Y)}{p(Y)}$.

**Lemma 1.** *Take $\pi$ a probability distribution on $\mathcal{A}$, then $\forall s \in \mathcal{S}$,*

$$\log p(s) \geq \mathbb{E}_{a \sim \pi(A)} \log p(s, a) - \log \pi(a)$$

*Proof.* Noting that $\log p(s, a) = \log p(s) + \log p(a|s)$:

$$\mathbb{E}_{a \sim \pi(A)} \log p(s, a) - \log \pi(a) = \log p(s) - \mathbb{E}_{a \sim \pi(A)} \left( \log \pi(a) - \log p(a|s) \right)$$

and $\mathbb{E}_{a \sim \pi(A)} \log \pi(a) - \log p(a|s) \geq 0$ (Kullback-Leibler divergence positivity). $\square$

**Lemma 2.** *Take $q$ a probability distribution on $\mathcal{Z}$, then $\forall (s, a) \in \mathcal{S} \times \mathcal{A}$,*

$$\log p(s, a) \geq \mathbb{E}_{z \sim q(Z)} \log p(s, a|z) - \log q(z) + \log p(z)$$

*Proof.* Noting that $\log p(s, a|z) + \log p(z) = \log p(s, a) + \log p(z|s, a)$ (Bayes Theorem)

$$\mathbb{E}_{z \sim q(Z)} \log p(s, a|z) - \log q(z) + \log p(z) = \log p(s, a) - \mathbb{E}_{z \sim q(Z)} \left( \log q(z) - \log p(z) \right)$$

and $\mathbb{E}_{z \sim q(Z)} \log q(z) - \log p(z) \geq 0$ (Kullback-Leibler divergence positivity). $\square$

**Lemma 3.** *Take $\pi$ a probability distribution on $\mathcal{A}$ and $q$ a probability distribution on $\mathcal{Z}$, then $\forall s \in \mathcal{S}$,*

$$\log p(s) \geq \mathbb{E}_{(a,z)\sim(\pi(A),q(z))} \log p(s,a|z) - \log \pi(a) - \log q(z) + \log p(z)$$

'

*Proof.* From lemmas 1 and 2, use the transitivity of the $\geq$ operator $\qquad\square$

Remark: the random variable $Z$ has been used here for readability. Taking $Z = (S^+, A^+)$ with $(S^+, A^+) \in \mathcal{S} \times \mathcal{A}$ and $p(Z) = p(S^+, A^+)$, the previous properties still hold.

### A.2 THE REWARD CONSTRAINT

Take $\beta \in \mathbb{R}^+$, $\gamma \in ]0,1]$, and $r : \mathcal{Z} \mapsto \mathbb{R}$ a reward function.

Note $R(z) = \frac{r(z)}{1-\gamma}$.

Assume that $p(A|s)$ (said the "optimal policy") is such that :

$$p(a|s) = \frac{\exp \beta \mathbb{E}_{z\sim p(Z|s,a)} R(z)}{\sum_{a'\in\mathcal{A}} \exp \beta \mathbb{E}_{z'\sim p(Z|s,a')} R(z')}$$

and define $K(s) = \log \sum_{a\in\mathcal{A}} \exp \beta \mathbb{E}_{z\sim p(Z|s,a)} R(z)$

**Lemma 4.** $\mathbb{E}_{z\sim p(Z|s,a)} \log p(s,a|z) \geq \mathbb{E}_{z\sim p(Z|s,a)} \beta R(z) - K(s) + \log(p(s))$

*Proof.*

$$\log p(s,a) = \log p(s) + \log p(a|s)$$
$$= \log p(s) + \mathbb{E}_{z\sim p(Z|s,a)} \beta R(z) - K(s)$$

From Bayes rule, we know that: $\forall z: \log p(s,a) + \log p(z|s,a) = \log p(z) + \log p(s,a|z)$. Then:

$$\mathbb{E}_{z\sim p(Z|s,a)} \log p(s,a|z) - \log p(z|s,a) + \log p(z) = \mathbb{E}_{z\sim p(Z|s,a)} \beta R(z) - K(s) + \log p(s)$$

Noting that:

$$\mathbb{E}_{z\sim p(Z|s,a)} \log p(z|s,a) - \log p(z) \geq 0$$

it comes that:

$$\mathbb{E}_{z\sim p(Z|s,a)} \log p(s,a|z) \geq \mathbb{E}_{z\sim p(Z|s,a)} \beta R(z) - K(s) + \log p(s)$$

$\qquad\square$

Remark: in the main text, $\bar{R}(s,z) = \mathbb{E}_{z\sim p(Z|s,a)} R(z) - \frac{1}{\beta}(K(s) - \log p(s))$ is said the "extended return".

Define now $\pi(A|s)$, a conditional distribution on actions, said the "current policy". Then:

**Lemma 5.**

$$\mathbb{E}_{a\sim\pi(A|s)} \log p(s) + \log \pi(a|s) \geq \mathbb{E}_{\substack{a\sim\pi(A|s) \\ z\sim p(Z|s,a)}} \beta \bar{R}(s,z) - \log p(z|s,a) + \log p(z)$$

*Proof.* Identify from lemma (3) $\pi(A) \equiv \pi(A|s)$ and $q(Z) \equiv p(Z|s,a)$. Then from lemma 4, it comes:

$$\log p(s) \geq \mathbb{E}_{\substack{a\sim\pi(A|s) \\ z\sim p(Z|s,a)}} \beta \bar{R}(s,z) - \log \pi(a|s) - \log p(z|s,a) + \log p(z)$$

which proves the formula. $\qquad\square$

The right term of the formula is said the (log)-evidence lower bound (ELBO) in the main text. It is constructed such that $\beta \bar{R}(s,z)$ is interpreted as the log likelihood of $(s,a)$ (even if it is, more strictly speaking, a lower bound of the log likelihood).

### A.3 OPTIMIZATION

The ELBO being bounded from above by $\log p(s)$, one can now define a (local) optimization criterion. Say $\Pi$ be a family of policies. Then the optimization criterion is:

$$\max_{\pi \in \Pi} \mathbb{E}_{\substack{a \sim \pi(A|s) \\ z \sim p(Z|s,a)}} \bar{R}(s,z) - \log \pi(a|s) - \log p(z|s,a) + \log p(z)$$

This criterion contains by construction two optimization objectives.

- A first objective is said the reward density-matching objective:

$$\max_{\pi \in \Pi} \mathbb{E}_{\substack{a \sim \pi(A|s) \\ z \sim p(Z|s,a)}} \bar{R}(s,z) - \log \pi(a|s)$$

It is constructed so that the log-probability of choosing action $a$ is proportional to the average reward observed at further stages of the dynamics.

- A second objective is said the regularization objective:

$$\max_{\pi \in \Pi} \mathbb{E}_{\substack{a \sim \pi(A|s) \\ z \sim p(Z|s,a)}} \log p(z) - \log p(z|s,a)$$

Taking a uniform prior distribution $p(Z)$ imposes for the policy to make the successor $Z$ to fit, on average, a uniform distribution.

