# OpenReview forum: "Occupy & Specify: Investigations into a Maximum Credit Assignment Occupancy Objective for Data-efficient Reinforcement Learning"
_ICLR.cc/2022/Conference — ICLR 2022 Submitted_

### Official Review · Reviewer_6SWm · 2021-11-01

**Correctness:** 3
**Technical Novelty And Significance:** 2
**Empirical Novelty And Significance:** 2
**Recommendation:** 3
**Confidence:** 4

**Main Review:**

Overall, I found the organization and clarity of the paper to be poor. Consequently, it wasn't clear to me what the theoretical contributions of the paper were. As far as I could tell, 2.1-2.6 were background and a refactoring of notation/existing ideas, and it wasn't clear to me how novel 2.7 is compared to similar methods in the literature. I provide some more detailed comments on how the clarity can be improved at the bottom.

The empirical evaluation is also lacking. In short:
- Missing baselines. The authors don't compare against other methods which aim to maximize state entropy.
- Poor reproducibility. Missing # of trials/seeds. Visually it seems there is some shading which suggests multiple seeds, but this information is absent. Weirdly, this error term disappears over average reward and cumulative rewards. There's per environment hyperparameters and the graphs seem to finish at arbitrary time steps, suggesting these results have been overly tuned to outperform the baselines. Code is not provided (although promised), and the method is not reproducible from only the details given in the paper.
- Missing analysis. Does the method improve state entropy or add to exploration? There's also no ablation study or any addition insight about the empirical properties of the algorithm.

I've scored my confidence as 4 because I am confident in my score, given the current state of the paper, however, I am not confident I have fully understood all the details of the paper.

Suggestions for clarity:
- $q_\pi$ notation is never defined. $K$ in section 2.4 is never defined.
- Section 2 should be split into background and a section which defines contributions. I'm not sure Section 2 needs so many subsections, and there is no explanation besides from the title, of what each subsection is meant to demonstrate.
- Too many recursive definitions. For example Eqn (3) is used solely to define $\beta \bar{R(s^+,a^+)}$, which is then replaced with $\tilde{log} p(s,a)$.
- It would be easier if any unconventional notation was defined in one place rather than scattered throughout section 2.
- ~ is used weirdly, for example Q(s,a) is "sampled" from the reward function above equation 4. Presumably this is meant to replace expectation but should be defined.
- Inconsistent notation. S,A are often used but are never defined. Are these meant to be the set of all possible state/actions, as defined by $\mathcal{S}$ and $\mathcal{A}$ in 2.1?
- Problem statement really needs to be split into at least 2, if not more, paragraphs.
- The full algorithm description should be moved from the appendix into section 3. The current algorithm description is also unclear in details. For example, "update the critic over all losses" is unclear which losses are being referred to.

Minor comments on clarity
- Given average return is undiscounted, I'm not sure average reward is an interesting metric to look at.
- Ordering environments by difficulty is subjective. Describing the environments in detail can be delayed to the appendix.
- Typo in 2.1 [0,1[ (should also be [0,1) if we're considering a discounted objective).
- It would be easier to follow if $S^+$ notation was introduced above the equation at the end of page 3.


**Summary Of The Paper:**

The authors introduce a RL method for optimizing entropy over states, as opposed to actions. The algorithm is tested on three MuJoCo domains and one classic control environment.

**Summary Of The Review:**

Poor clarity makes the contributions hard to follow, along with weak empirical analysis puts the paper below the bar for acceptance.

---

> ### Author Response · Authors · 2021-11-18
> **response to reviewer 6SWm**
>
> Thanks for the thorough reading and complementary remarks.
> Some questions about the mathematical notations have been treated in the other responses so I do not develop here.
> Packing everything under the 9 pages limit was made at the price of putting the detailed algorithm in the appendix. Both reviewers suggested instead putting the algorithm detail in the main body and displacing the gym task details in the appendix, which should be done in the new version.
>
> Now regarding the numerical/empirical evaluation of our algorithm versus baselines:
>
> First to be noticed is that the general justification of the approach is the data efficacy. The baseline algorithms are found to be too greedy and not enough exploratory in 2 tasks over 5. Those 2 tasks (continuous mountain car and swimmer) were supposed to be simpler (low dimension) but revealed to be more challenging for the baselines.  The three other tasks put our algorithm in the top 2 regarding the total (undiscounted) return and top1 regarding the cumulative rewards, which proves it to be safe and stable.
>
> Now regarding your specific remarks:
> - The comparison is made with baselines that contain an exploratory term (that is an entropy bonus on actions) but not an entropy bonus on states. We are not aware of any "intrinsic reward" (pseudo counts or learning improvement) implementation in the continuous action case. The many implementations of intrinsic reward approaches concentrate on the discrete action/sparse reward case (like in Atari games). Those environments are very relevant to our method, but it would be longer to develop and tune for it would require an explicit auto-encoder. This is the subject of future research. What we illustrate here is the fact that implementing a state entropy bonus is possible in environments with inputs and outputs being multidimensional real vectors. This was not obvious at first stake and we show it to be relatively simple to implement.
> -  the number of seeds varies from one experiment to the other, and this was not precise in the text. This is corrected in the new version,  with 5 seeds per experiment except for the humanoid case. The code will be made available at publication time and everything should be testable and reproducible. A light parameter tuning was made for each experiment, including the size in hidden layers, $\beta$, $\lambda$ and $\gamma$. The $\gamma$ parameter and size of layers is shared with every method. The $\beta$ parameter is shared only with the sac algorithm. The $\lambda$ parameter is specific to MaCAO. The parameter optimization was done by grid search (except for the layer size that was set once in function of the dimensionality of the problem). The optimization on $\gamma$ was beneficial for all methods. The optimization on $\beta$ impacts also the performance of SAC,  which may introduce a bias in the comparison. This will be detailed in the final version.
> - the method was implemented in another (grid world) setup, but there is not enough space here to detail and present this alternate implementation, over which the pure exploration capabilities are easier to illustrate.  The continuous action case, as proposed in the Gym continuous action suite contains many "trap" states that conduct to an abrupt ending of the episode.  In a pure exploration setup, those states will not be avoided and early ending will be undergone most of the time. The exceptions are the continuous mountain car, the swimmer (and the pendulums and reacher that are "too easy" for consideration). We tested a pure exploration setup in the continuous mountain car and interestingly found that "wide exploration" is enough incentive to reach the terminal state in a significant proportion of trials (but less systematically than when the end-of-episode bonus is considered).  The full exploration analysis should be developed in a full paper but will not be developed here for conciseness.
>
> Other remarks:
> - The novelty of sections 2.4-2.6 is the introduction of the extended reward trick and reformulation of variational optimization over occupancy distributions (rather than time-resolved trajectories). The full detail is in the response to reviewer yV1o.
> - The average reward provides an interesting insight in both the bipedal and the humanoid case. It shows that maxent on state adds to the baseline methods in finding ways to improve the dense reward by exploring more risky behaviors. This corresponds to the attraction toward states that are both more rewarding and more risky (in particular greater risk of falling down when moving faster). The risk/benefit compromise is displaced toward higher risk in the maxent-on-state case, and this is an unexpected result of our approach.

---

> > ### Comment · Reviewer_6SWm · 2021-11-22
> > **Thank you for the response**
> >
> > Thank you for the response. I don’t currently feel as if the changes are sufficient for an increase in score, but they do help improve the paper. I may adjust my score after discussion with the other reviewers.
> >
> > Regarding baselines, consider [1,2].
> >
> > [1] Lee, Lisa, et al. "Efficient exploration via state marginal matching." (2019).
> > [2] Seo, Younggyo, et al. "State entropy maximization with random encoders for efficient exploration." (2021).

---

> > > ### Author Response · Authors · 2021-11-24
> > > **Papers comment**
> > >
> > > Hi
> > > Thanks for the scan of the bibliography. This appeals few comments.
> > > - Lee et al is already cited in our paper. The method they propose (mixture of mixtures) was not provided with an actual python implementation, and is not compared with standard benchmark in their paper (and I am not aware of a follow-up paper). My feeling is that they adapt their method and their algorithm to the various tasks they consider. This makes their results difficult to reproduce from scratch.
> > > - We effectively missed the "Seo et al" reference. Sadly they test their algorithm on the Deepmind suite rather than the gym, making on-the-fly comparison impossible. This one should clearly be cited in a final version (if ever) and seems quite straightforward to implement (we effectively tried this projection trick but it revealed to be less efficient in our case than the baseline kernel-based approach)

---

### Official Review · Reviewer_yV1o · 2021-11-02

**Correctness:** 3
**Technical Novelty And Significance:** 1
**Empirical Novelty And Significance:** 2
**Recommendation:** 1
**Confidence:** 3

**Main Review:**

### Quality
* #### Weaknesses
    * ##### Major
        * While numerous mathematical statements are made throughout the paper, some are already well-established in the reinforcement learning literature (the chain of statements ending Section 2.2 should all hold with equality and dates back to the successor representation[1]) and others are simply stated as fact without proof, although surrounding text would suggest that the authors are claiming them as technical contributions of this work. While the introduction of variational approximations leads to lower bounds that resemble those shown in the paper, the various notations and distributions introduced make it difficult to confirm their validity right away. The authors should (at a minimum) include proofs for novel technical contributions of this work and further include proofs for supporting results that may help the reader understand the nature of their contributions.
        * In Equations 9 and 10, (my understanding is that) the included log-likelihood ratio is between the variational approximation of the behavior policy visitation and that of the optimal policy. It seems rather likely that this ratio may suffer from numerical errors (namely, division by zero) when the exploratory behavior policy visits a region of the state-action space not covered by the optimal policy. Could the authors comment on why this is not an issue? Or are the authors making some kind of absolute continuity assumption?
        * Ultimately, this paper is deeply concerned with addressing the challenge of exploration. Can the authors comment on why this formulation under probabilistic inference is reasonable given recent work which shows how the approximations needed for these methods to be applied in practice fail to pass even basic sanity checks? [2]
	* ##### Minor
		*

### Clarity
* #### Strengths
    * The clearest element in the paper is Algorithm 1, which at least conveys the generic structure of what the authors are trying to achieve. Unfortunately, everything leading up to Section 3 is incredibly unclear stemming from a variety of sources including non-standard mathematical notation, undefined jargon/terminology, and (to a lesser extent) grammatical errors. The authors have wasted a page describing details of widely-known Mujoco environments that could instead be allocated to adding clarity to Sections 2 and 3, which seem to articulate the bulk of the contribution.
* #### Weaknesses
    * ##### Major
        * The mathematical notation used throughout the paper is terribly difficult to parse and inconsistent with the broader reinforcement-learning literature. As a concrete example, Section 2.3 contains undefined notation (\mathcal{T}\pi(s)), ambiguous notation (s... notation, s+ appears to denote a state and yet is later said to denote a sequence of future states but then used as a normal state in Equation 3), and nonsensical notation (the second paragraph of Section 2.5 has Q(s,a) \sim \frac{r(s+,a+)}{(1-\gamma)}, which is entirely vacuous without being defined beforehand). The authors seem to jump back and forth between states indexed by time vs. not.
        * The authors make a bad habit of introducing assumptions unnecessarily. In Section 2.4, if the first assumption on Boltzmann policies is made, then why is the second assumption on the existence of K(s) needed as well? The statement should hold under the first assumption and K(s) can be written out explicitly. Why is the first assumption of Section 2.5 needed as if it re-defines the optimal policy? Unless the policy class over which the argmax is taken is a subset of all policies, this is the definition of the optimal policy for a MDP.
        * The authors introduce their own terminology without any formal definitions or visualizations to assist the reader in understanding what the terms are meant to convey. As a result, the first paragraph of Section 2.7 reads vacuously without knowing what is meant by "pathways", "conducts", "adverse", through", or "width."
        * The paper has several grammatical errors throughout. While a small handful of these would not constitute a major weakness (and, in fact, could be easily enumerated and corrected), there are enough of them to impact readability of the paper.
	* ##### Minor
		* The use of the phrase "probability matching" in this paper is misleading since it is widely associated with Thompson sampling. Distribution matching would be a more accurate phrase for what the author is trying to convey.


### Originality
* #### Weaknesses
    * ##### Major
        * The paper lacks a section dedicated to providing readers with an overview of related prior work. Consequently, it is difficult to precisely identify how the authors' variational formulation extends or expands beyond some of the prior work mentioned in brief at the top of page 2.
	* ##### Minor
		*

### Significance
* #### Weaknesses
    * ##### Major
        * The authors haven't included details of how many random seeds are used to generate the results in Figure 1; consequently, I'm skeptical of their significance and reproducibility. That said, just examining the average return column (which is the standard metric), it seems that the proposed approach only leads to meaningful performance improvements in simpler control problems, while results in Swimmer and Humanoid are only on par with SAC.
        * Simpler experiments in domains where things can be concisely and cleanly visualized would go a long way towards driving home the central claim that this approach facilitates improved exploration in practice.
	* ##### Minor
		*

# References
1. Dayan, Peter. "Improving generalization for temporal difference learning: The successor representation." Neural Computation 5, no. 4 (1993): 613-624.
2. O'Donoghue, Brendan, Ian Osband, and Catalin Ionescu. "Making Sense of Reinforcement Learning and Probabilistic Inference." In International Conference on Learning Representations. 2019.


**Summary Of The Paper:**

# Summary & Contributions
* This paper examines a variational approach to reinforcement learning, leveraging occupancy measures over previously visited and future state-action pairs in order to address the exploration challenge.
* The author propose a variational approximation to the so-called "conditional occupancy" of the current behavior policy denoting the distribution over future state-action pairs. The approximation is used to induce broader state visitation from the behavior policy, coupled with a standard actor-critic algorithm.
* Empirical results show considerable performance gains in simpler continuous control problems and matching performance against baseline methods in more high-dimensional control tasks.

**Summary Of The Review:**

# Final Remarks
* As evidenced by my main review, I have struggled to identify strengths of this paper, in large part because the precise nature of the technical contribution and how it improves upon prior work is unclear to me.
* Still, the empirical results are at the level of a preliminary workshop paper and more experimentation seems warranted across a broader set of environments to confirm the utility of the proposed approach.
* The clarity issues raised above suggest that the paper requires substantial revision in order to more clearly convey the contributions and algorithms to readers.
* For now, my score is quite low though I would be happy to raise it in light of suitable clarifications from the authors.

======= Post Rebuttal =======

I thank the authors for their response but the lack of clarity around the approach itself and originality as well as issues with the writing style point to substantial revisions that are needed before the submission is ready for publication.

---

> ### Author Response · Authors · 2021-11-17
> **response to reviewer yV1o**
>
> Thanks for the Dayan (1993) reference and very thorough reading that is much appreciable. The conditional occupancy developed in the paper is indeed an instance of the successor representation/distribution.
>
> Now regarding your main remarks:
> - Regarding the mathematical notations: the few mathematical notations introduced in the paper are there to provide a sense of "further away" or "indefinite future state". For instance, the + superscript is there to visually identify $s^+$ as a state that is visited "further away" after s, while $s'$ generally represents the state that immediately follows s in the sequence of visits. Then the ... dots in conditional probabilities were introduced because the $p_\pi$ and $\rho_\pi$ conditional distributions were visually indistinguishable. the triple dots simply illustrate a temporal gap between the main variable and the conditional variable. Few other notations ($\mathcal{T}_\pi(s)$) are indeed not introduced but this will be corrected.
> Then, $Q(s,a) \sim \frac{r(s+,a+)}{(1-\gamma)}$ is a shorthand for the formula at the top of page 4. This is also the case for the next formula.
> - The time index is put when trajectories (sequences of states) are considered and bypassed when considering a sample of the occupancy distribution.
> - Now let us come to the main contribution. Everything that is introduced in sections 2.4 and 2.5 is there to give a sense of interpreting rewards as probabilities. Everyone knows that the Boltzmann formula transforms a reward into a probability over actions. Now the trick is to extend the Boltzmann distribution toward a distribution over state and action pairs.  This is the meaning of the "extended reward". Once this is done, the optimization becomes probability (or density) matching on occupancies. This recasting of softmax/Bellman optimization into occupancy matching provides, as shown by formula (4) and (5), significantly simpler objective functions (when comparing to matching on trajectories).
> Then,  it also seems obvious that formulas (4) and (5) are not entirely satisfactory: the average over rewards translates into average over log-density samples, that have actually depend on (s+,a+), though it is not explicit in the formula.
> This conducts to considering a (reverse) conditional occupancies instead : rho(s,a|...s+,a+), with the dependency on (s+,a+) clearly expressed.  The extended reward now becomes (formally) a log-likelihood, the future (state, action) pair becomes a data model, which allows to reframe the full optimization setup into bayesian variational optimization.
> All the elements of variational inference are thus clearly outlined, though the elements that compose it are not the usual ones.
> (namely, the causal dependency is reversed with regards to classic inference). This provides a certain number of classic properties, most of which that the maximum exists (provided  $\rho^*(s^+,a^+)$ is nonzero on its support and the rewards non-singular, which is commonplace). It also provides a way to follow two objectives simultaneously. It is probably not emphasized enough in the paper, but from now on, there was no proper (or principled) way to both follow a maxent-on-states objective (that is exploration) and reward-seeking objective. The many "intrinsic reward" setups consider the intrinsic reward should progressively vanish as the optimization progresses. This is not the case here. If the rewards collected express an occupancy over the states (say objective A), there is no necessity for the prior $\rho^*(s^+,a^+)$ to express the same objective (say objective B). Define for instance the prior as uniform over the states pushes the policy toward exploring new states (rarely visited). The exploration and the reward seeking are thus pushing the policy into different directions, and this conducts toward an equilibrium. In the uniform prior case, the prior plays the same role as the regularizer in VAE, which allows to avoid local minima and overtraining.
>
> - Regarding your remark about equations 9 and 10, the prior is uniform over the state space in our implementation and is thus considered as a constant.
>
> - the paper of O'Donoghue et al mostly refers to the approach of  Abdolmaleki  et  al.,2018 and Haarnoja et al., 2018, which is a maxent-on-action variational formula. Introducing maxent-on-states in a variational formula, as we do,  provides a way to address some of the shortages listed in the paper. The "sanity check" we propose here is the continuous mountain car, whose dense rewards (cinematic cost) are inconsistent with the distal (end of episode)  bonus. This is conceptually equivalent to the "deep sea" exploration problem in grid-worlds. Our algorithm solves the problem, while the baselines do not.

---

### Official Review · Reviewer_FQij · 2021-11-02

**Correctness:** 3
**Technical Novelty And Significance:** 2
**Empirical Novelty And Significance:** 2
**Recommendation:** 3
**Confidence:** 3

**Main Review:**

First sentence "to deal" -> "dealing" . "Open-ended learning" is not defined, I don't know what this means. What does it mean to "carry out" a model?  Page  " including the curiosity drives" -> "including curiosity drives". Some citations are missing parentheses. The next sentence doesn't make sense. The authors here cite the "E3D" algorithm and "MaxEnt" with which they compare in the empirical section, but for somebody not familiar with these algorithms, the paper should at least briefly describe how these are different from their proposal for these comparisons to make sense. "the dynamics of the environment is Markovian" -> "the dynamics of the environment are Markovian", "The dynamic programming objective is described like:", is that a definition rather? Page 3, what is a "set point"?  "to seek for" -> "to seek"? "so as to match" -> "to match".  The paper uses its own language which is not formal or universal, and quite confusing, and introduces many shorthands on top of shorthands. It uses capital letters and lower case letters arbitrary at times, although I assume the regular rule of capital meaning random variables is the intention. What is a "value sample"? Waht does the "future of s_t" mean? What is K? it is not introduced, I assume the log sum exp/partition/normalization function? In section 2.4 equations do not easily from one from the other, leaving the readers confused at times, and having to do these substitutions themselves.  Why is the $\rho(s)$ appearing in equation 3. Where does it come from? When you say something contains "a guess" about something else, what does that mean in a standard concrete mathematical way? What does $\bar$ over a function mean? What does $\tilde$ over a function mean? At the end of page 4., the author say the next objective function is the one considered by Soft-actor critic, but that is not obvious, SAC works in policy space, not over the space of occupancy maps. What is "a complement term". Using again $q_\pi$ not to mean the action-value function, but a stationary distribution, is very confusing. The paper jumps to the variational perspective, but does not explain this interpretation and why it is helpful or provably useful.  Above section 2.7, "at the light" -> "in light", "the progress of the training" -> "training progress". Section 2.7, "goal pathways as modes of the objective occupancy", what does that mean mathematically, what is the \emph{objective} occupancy. What is "a through" (noun)? What does the "width" at which it is influential on the number of pathways" mean? Next sentence confuses upper case S, A with lower case s,a and reverses them.  What is the "objective credit assignment"? it's not clear what this means. On page 6, there is a claim that exploration in this way of entropy maximization helps the policy converge faster to the optimum, but it is not supported by anything (citation, or results inside the paper), "for the implies to concurrently" -> "for this implies to concurrently". What does "overfitting" mean here? we have results in policy optimization explaining slow convergence due to flat regions of the optimization landscape. "state transition vanish during the optimization...", what vanishes? Page 7, Sutton(1988) is missing citation parentheses. The last sentence of that paragraph, is probably a hypothesis explored in this paper, but do we know for certain this is the case? The exact algorithm is not described in the paper, the authors pointing to the appendix for the most important aspect of the paper, such as how does one compute the log-occupancy? It also mentions the use of a "kernel-based density estimation method" which appears to be fundamental, but this is not explained or discussed. The algorithm implementation or pseudo-code is also not given in the paper. "Results" section, "several benchmark environments", which? and why those? "expressed as average return" -> ""expressed as average return". Page 8, "comparing with the SAC" -> ""comparing with SAC". Page 9, contains an interesting description of the games, showing insight. It ends with "this ripple pattern is not contain in the reward signal", what does that mean?

**Summary Of The Paper:**

The authors address the problem of exploration in reinforcement learning. They suggest applying the maximum entropy principle over the state space and try to maximize the make the occupancy of the policy as entropic as possible while still optimizing the original goal of maximizing cumulative reward. They claims include an algorithm that does this using a replay buffer and off-policy learning, which improves sampling efficiency, in sparse and dense reward settings, in classic motor control benchmarks.

**Summary Of The Review:**

The paper addresses the exploration-exploitation trade-off proposing the maximum entropy principle over the state-space as a solution to this problem. Although a promising avenue, many parts of the paper are confusing, using inside jargon, and decreasing the overall clarity. Some parts of the paper are not supported by results, and the main methods used are referred to the appendix.

---

> ### Author Response · Authors · 2021-11-16
> **reponse to reviewer FQij**
>
> Thanks for the efforts in reading and commenting the paper. There are many short and unsorted comments in your review. I cannot address all of them here, but all of them are valuable and should be considered in the last version.
> - concerning your remark about the maxent (on states) algorithms : maxent means optimize the policy so as the states visited under policy pi are maximally variable, ideally uniformly visiting all possible states. In contrast to pure maxent, our approach provides a way to combine the maxent objective with a reward maximization objective, under a variational inference perspective.
> - The SAC objective was initially formulated as a KL divergence on trajectories, not states. The formula (5) is a recasting of the original SAC formula under the "occupancy" setup.
> - the maximization of the entropy on states introduces a bias toward visiting the states that are rarely encountered, helping to collect more diverse data and avoiding overfitting. the algorithm is never claimed to converge faster to the optimum (indeed it should never reach the optimum!), but expected to be data efficient, and avoid being trapped in local optima.
> - the choice of the specific benchmark environments --> the algorithm is implemented in the continuous action setup, which explains the use of mujoco and box2D environments. The specific selection of the tasks is justified in the text (namely tasks with and/or without end-of-episode bonus, consistent and/or inconsistent dense rewards)
>
> Diverse remarks :
> - open-ended learning --> unbounded environments
> - What does it mean to "carry out" a model? --> either engineer or train a model
> - the use of capital letter is not arbitrary. The capital letter is a canonical notation for random variables, while the script letter refer to realizations. Typically, p(X|y) is a shorthand for p(X|Y=y) the conditional distribution of random variable X, knowing y.
> - value sample: shorthand for saying the (action)-value can be here expressed as an expectation on rewards
> - the future of s_t is any s_t', t'>t
> - "guess" stems here for "realization"
> - the \bar and \tilde notations are not satisfactory and will be changes in the final version
> - $\rho(s)$ (in eq.3) is the marginal occupancy, defined in eq.1
> - q_pi stems for a variational occupancy that is optimized through variational inference. It does not relate to Q (the action-value function)
> - a through --> a trough (typo)
> - objective occupancy --> target occupancy
> - "width" --> all the pathways heading to (s+,a+) are here interpreted as the branches of a tree...
> - "objective credit assignment" --> occupancy credit assignment
> - state transitions vary during the optimization --> the state transitions do not depend on pi so the differential is zero
> - the "kernel-based density estimation method" used here is the "nearest neighbour" implementation from sklearn (this will be made more explicit in the paper)
> - the ripple pattern is not contained --> emerges from the constraints

---

> > ### Comment · Reviewer_FQij · 2021-11-29
> > **Thank you for the response.**
> >
> > Thank you for the response. I will maintain my original score.

---

### Decision · Program_Chairs · 2022-01-20

**Decision:**

Reject

**Comment:**

While the main idea of the paper (using a Max-Ent objective on the states of an MDP) was considered interesting, all reviewers raised the problem of clarity of the paper which needs to be drastically improved. While the writing could be improved by the revsion, these concerns could also not be fully alleviated by the rebuttal of the authors. The reviewers agreed that the paper needs rewritting in order to clarify the contribution before the paper can be published.